# Advanced Resistance Studies Identify Two Discrete Mechanisms in *Staphylococcus aureus* to Overcome Antibacterial Compounds that Target Biotin Protein Ligase

**DOI:** 10.3390/antibiotics9040165

**Published:** 2020-04-06

**Authors:** Andrew J. Hayes, Jiulia Satiaputra, Louise M. Sternicki, Ashleigh S. Paparella, Zikai Feng, Kwang J. Lee, Beatriz Blanco-Rodriguez, William Tieu, Bart A. Eijkelkamp, Keith E. Shearwin, Tara L. Pukala, Andrew D. Abell, Grant W. Booker, Steven W. Polyak

**Affiliations:** 1School of Biological Sciences, University of Adelaide, South Australia 5005, Australia; andrew.hayes@unimelb.edu.au (A.J.H.); jiulia.satiaputra@perkins.uwa.edu.au (J.S.); l.sternicki@griffith.edu.au (L.M.S.); ashleigh.paparella@einsteinmed.org (A.S.P.); zikai.feng@utas.edu.au (Z.F.); bart.eijkelkamp@flinders.edu.au (B.A.E.); keith.shearwin@adelaide.edu.au (K.E.S.); grant.booker@adelaide.edu.au (G.W.B.); 2School of Physical Sciences, University of Adelaide, South Australia 5005, Australia; kwangjun.lee@adelaide.edu.au (K.J.L.); bea_blanco_r@hotmail.com (B.B.-R.); william.tieu@sahmri.com (W.T.); tara.pukala@adelaide.edu.au (T.L.P.); andrew.abell@adelaide.edu.au (A.D.A.); 3Centre for Nanoscale BioPhotonics (CNBP), University of Adelaide, Adelaide, SA 5005, Australia; 4Institute of Photonics and Advanced Sensing (IPAS), School of Biological Sciences, University of Adelaide, Adelaide, SA 5005, Australia

**Keywords:** antimicrobial resistance, Gram-positive bacteria, *Staphylococcus aureus*, advanced resistance studies, biotin, biotin protein ligase, BirA, novel antibacterials

## Abstract

Biotin protein ligase (BPL) inhibitors are a novel class of antibacterial that target clinically important methicillin-resistant *Staphylococcus aureus* (*S. aureus*). In *S. aureus,* BPL is a bifunctional protein responsible for enzymatic biotinylation of two biotin-dependent enzymes, as well as serving as a transcriptional repressor that controls biotin synthesis and import. In this report, we investigate the mechanisms of action and resistance for a potent anti-BPL, an antibacterial compound, biotinyl-acylsulfamide adenosine (BASA). We show that BASA acts by both inhibiting the enzymatic activity of BPL in vitro, as well as functioning as a transcription co-repressor. A low spontaneous resistance rate was measured for the compound (<10^−9^) and whole-genome sequencing of strains evolved during serial passaging in the presence of BASA identified two discrete resistance mechanisms. In the first, deletion of the biotin-dependent enzyme pyruvate carboxylase is proposed to prioritize the utilization of bioavailable biotin for the essential enzyme acetyl-CoA carboxylase. In the second, a D200E missense mutation in BPL reduced DNA binding in vitro and transcriptional repression in vivo. We propose that this second resistance mechanism promotes bioavailability of biotin by derepressing its synthesis and import, such that free biotin may outcompete the inhibitor for binding BPL. This study provides new insights into the molecular mechanisms governing antibacterial activity and resistance of BPL inhibitors in *S. aureus*.

## 1. Introduction

The need for new antimicrobial agents to combat the growing threat of drug resistance is currently not being met, with only five novel classes of antibiotics introduced since 2000 [1,2]. Strains that are now resistant to multiple classes of clinically approved antibiotics have been reported for *Staphylococcus aureus, Enterococci* species and *Mycobacterium tuberculosis* amongst others [3]. Without new antimicrobial agents, the inability to treat resistant microbial infections is predicted to result in 10 million deaths annually by 2050 [4]. Hence, there is a desperate need for new products with novel mechanisms of action that are not subject to existing resistance mechanisms. Precious few novel antimicrobial agents are currently in the drug development pipeline. One promising new drug target is biotin protein ligase (BPL) for which new potent, selective inhibitors are being developed (reviewed in [5,6]). In this report, we examine one such inhibitor that is active against drug-resistant *S. aureus* [7] and probe mechanisms of action and potential resistance.

In certain bacteria, such as clinically important *S. aureus*, BPL is a bifunctional protein with two important roles; (1) enzymatic biotinylation of biotin-dependent enzymes, and (2) biotin-dependent repression of certain target genes implicated in biotin homeostasis. This bifunctionality makes BPL the key master regulator of all biotin-mediated metabolic processes in *S. aureus* and a promising new antibiotic target [6]. *S. aureus* BPL (*Sa*BPL) catalyzes the covalent attachment of a biotin prosthetic group onto two key biotin-dependent enzymes, namely acetyl-CoA carboxylase (ACC) and pyruvate carboxylase (PC). Protein biotinylation occurs through two partial reactions. Biotin is first ligated to ATP yielding the reaction intermediate biotinyl-5′-AMP (Figure 1). BPL then employs the labile biotinyl-5′-AMP to attach the biotinyl moiety onto a single target lysine residue present in the active site of ACC or PC [8]. The appended biotin is necessary for the biotin-dependent enzymes to fulfill their important metabolic roles. ACC catalyzes the first committed step in fatty acid synthesis, whilst PC replenishes the tricarboxylic acid cycle with oxaloacetate. ACC and BPL have been identified as essential gene products for growth in genetic knockout studies performed on *S. aureus* [9,10,11,12] and, consequently, have been the targets of drug discovery programs. The antibacterial efficacy of ACC inhibitors has been demonstrated in vivo (reviewed in [13,14]) and BPL inhibitors have shown efficacy in vitro against *S. aureus* and *M. tuberculosis* [15,16,17,18,19]. Whilst dispensable for growth in culture, PC has been shown to be an important virulence factor in bacteremia models in mice and nematodes [20,21]. As protein biotinylation is essential for the activity of ACC and PC, pharmacological inhibition of *S. aureus* BPL (*Sa*BPL) is proposed to impact bacterial viability and pathogenesis due to the disruption of both enzymes (reviewed [6]).

*Sa*BPL also regulates the transcription of three genetic elements, namely the biotin biosynthesis operon (*bioO*), a gene encoding the substrate-specific subunit of the biotin transport protein BioY, as well as a putative fatty acid ligase operon *yhfT-yhfS* [22,23]. Recent in vivo characterization in *S. aureus* has revealed a hierarchy in the control of these genes by *Sa*BPL. The biotin biosynthesis operon was shown to be the most responsive to treatment with exogenous biotin, being repressed by 10 nM biotin 15 min post-treatment [23]. Transcripts for *bioY* and *yhfT-yhfS* persisted longer after biotin treatment than the biotin biosynthesis genes. It is proposed that this mechanism may assist with metabolic adaptation of the bacteria to their environment, as *de novo* biotin biosynthesis is an energetically expensive process that can be bypassed when exogenous biotin is readily available for import. This was supported experimentally where the growth rate of *S. aureus* was enhanced in media containing exogenous biotin, even though the bacteria were capable of biotin synthesis [23]. We propose that this ability is important during infection as *S. aureus* occupies a variety of niche microhabitats with varying biotin availability [23]. These findings also suggest that pharmacological induction of transcriptional repressor activity may lead to biotin starvation through the dual actions of reduced *de novo* biotin synthesis and reduced biotin import. Hence, small molecules that serve as co-repressors are beneficial in the design of antibacterials that target *Sa*BPL.

Structural biology has greatly assisted in understanding the bifunctional activity of *Sa*BPL. The protein contains an N-terminal helix-turn-helix domain necessary for DNA binding (Appendix A; reviewed in [8,24]). When the cellular demand for biotin is low, and biotinyl-5′-AMP is not required for protein biotinylation, this reaction intermediate also serves as a co-repressor by inducing protein homodimerization, a pre-requisite for high-affinity DNA binding to the operator sequences and repression of the target operons and genes [25,26]. A series of well-characterized conformational changes occur in *Sa*BPL to facilitate this protein dimerization [27]. Following the binding of biotin, residues 118–129 (known as the biotin-binding loop) undergo a disordered-to-ordered transition that creates a pocket that allows binding of ATP. An adenosine-binding loop, formed by residues 224–228, also undergoes structural changes to stabilize the binding of the adenylate moiety [27]. Further allosteric conformational changes occur to loops located in the dimerization interface upon ligand binding. These include the ordering of loops contained between amino acids 142 and 150 (extending an alpha-helix by three residues) and amino acids 198–201 [27]. Together these structural rearrangements facilitate the transition from monomer to homodimer. Arg122 and Phe123 in the biotin-binding loop stabilize the dimer by interacting directly with the side chain of Asp200 in the opposing subunit (Appendix A) [26]. Homodimerization appropriately aligns the two N-terminal DNA binding domains to facilitate their interaction with DNA (Appendix A) [27]. Protein dimerization has been reported to be induced by chemical analogs of biotin and biotinyl-5′-AMP [15,26,28,29].

Several groups have reported on the development of anti-BPL compounds with antibacterial activity [15,16,17,18,19,30,31,32,33,34]. Bisubstrate analogs that occupy both the biotin and ATP-binding pockets and mimic the mode of binding adopted by biotinyl-5′-AMP have been particularly promising. Non-hydrolyzable analogs of biotinyl-5′-AMP have been generated by replacing the labile phosphoanhydride linker connecting the biotin and AMP moieties with alternative bioisosteres. These new inhibitors exploited intermolecular interactions with the bacterial BPLs that are absent in the human enzyme, to provide necessary selectivity. Recently, the bisubstrate analog biotinyl-acylsulfamide adenosine (BASA, Figure 1) was reported with potent anti-*S. aureus* activity (minimal inhibitory concentration (MIC) RN4220 = 0.125 µg/mL, ATCC 49775 = 0.25 µg/mL) and necessary in vitro selectivity over the human equivalent (*K_i_ Sa*BPL = 2.4 nM, *K_i_* human BPL >10 μM; >4000-fold selectivity) [7]. A full description of the synthesis of BASA and its chemical characterization has already been described [7]. A crystal structure of *Sa*BPL in complex with BASA revealed that the compound adopted the same mode of binding and induced the same conformational changes necessary for DNA binding and repressor activity as biotinyl-5′-AMP (Appendix A) [7]. In the current study, the cellular mechanisms that contribute to the antibacterial efficacy of BASA were explored. Two discrete resistance mechanisms were also discovered using advanced resistance studies and whole-genome sequencing analysis. This study provides new insights into the mechanism of action of anti-BPL compounds and possible resistance mechanisms to a novel class of antibiotics.

## 2. Results

### 2.1. Biological Evaluation of BASA

The antibacterial spectrum, therapeutic window and mechanism of action of BASA were first established using a series of microbiological and cellular assays. Antimicrobial susceptibility assays performed against a panel of clinical isolates of *S. aureus* (*n* = 23) revealed that BASA was active against both methicillin-sensitive and resistant strains with minimal inhibitory concentration (MIC) measurements ranging from 0.125 to 0.5 µg/mL (Table 1). There was no change to the MIC when the antimicrobial susceptibility assay was performed in growth media supplemented with 10% fetal calf serum but it did increase eight-fold with 20% serum, suggesting slightly reduced efficacy in the presence of serum fatty acids (Table 1). This potent antibacterial activity was restricted to Staphylococci as other bacterial pathogens were insensitive to BASA including Gram-positive *Streptococcus pneumoniae,* and Gram-negative *Escherichia coli* and Enterobacter species. *Mycobacterium tuberculosis* was the only other bacterium susceptible to BASA (MIC = 55 µg/mL, 100 µM), albeit with weaker potency than *S. aureus*. Cytotoxicity testing against two human cell lines HepG2 (liver-derived) and HEK293 (kidney-derived) showed both were insensitive to BASA at concentrations up to 250 μg/mL (Table 1, Appendix A), providing a >100-fold therapeutic window for *S. aureus*. Finally, concentration and time-dependent kill kinetics experiments were performed by viable cell counts following 24 hours of treatment with BASA at one, two and four times the MIC. Growth ceased at all concentrations, however, killing was not observed consistent with BASA functioning as a bacteriostatic antibacterial (Appendix A). The bacteriostatic activity of BASA implies that host-induced killing and/or combination therapy may be required for in vivo efficacy. Consequently, synergy assays with other clinically used antibacterial agents were performed using a checkerboard titration assay [35]. Of the eight antibacterial agents tested, none were antagonistic with BASA, and synergy was detected with both methicillin and streptomycin as determined by a fractional inhibitory concentration (FIC) of <0.5 (Appendix A) [35,36]. Together these biological data establish BASA as a promising new antibacterial agent.

### 2.2. Mechanism of Action of BASA

Three lines of evidence were consistent with BASA acting through direct binding to the BPL target. Firstly, the compound showed reduced antibacterial activity against a strain of *S. aureus* engineered to overexpress *Sa*BPL [19]. The increased concentration of recombinant BPL in the bacteria elevated the MIC by eight-fold over an isotypic control strain (Appendix A). This response was specific for BPL inhibition with a similar increase observed with another chemical analog of the reaction intermediate, biotinol-5′-AMP [32], but not the unrelated β-lactam amoxicillin (Appendix A). In further support, increased concentrations of the BPL substrate biotin in the growth media also reduced the antibacterial activity of BASA. Super-physiological concentrations of 1 μM (i.e., >1000-fold serum concentrations) of exogenous biotin increased the MIC of BASA by 64-fold (Appendix A). Finally, we demonstrated that BASA functions as a co-repressor in vivo with a potency equivalent to that of biotin. Electrophoretic mobility shift assays (EMSA) were first performed and confirmed that BASA could promote *Sa*BPL binding to DNA in vitro through the same mechanism utilized by the natural co-repressor biotinyl-5′-AMP (Appendix A). Here, the *Sa*BPL-BASA and *Sa*BPL-biotinyl-5′-AMP complexes were compared for binding to both the *bioO* and *bioY* promoter sequences. Prior to EMSA analysis, unliganded (apo) *Sa*BPL was incubated with either BASA, or biotin and MgATP to allow the formation of biotinyl-5′-AMP. Equipotent binding was observed for the two complexes to both the *bioY* and *bioO* sequences (Appendix A). Having established the DNA binding activity of the *Sa*BPL-BASA complex in vitro, the co-repressor activity of BASA was then investigated using transcriptome studies on whole cells. *S. aureus* NCTC 8325 was grown in biotin depleted media until log phase and then treated with 10 nM of either biotin or BASA. Cells were also treated with 3.9 μM of BASA, a level four times the MIC. Cells were harvested at 0, 15, 30 and 90 min post-treatment and RNA extracted for qRT–PCR analysis to measure the levels of the *bioD* (the first gene in the biotin biosynthesis operon) and *bioY* (biotin transporter gene) transcripts (Figure 2). *S. aureus* treated with 10 nM biotin displayed a 111-fold decrease in *bioD* expression at 15 min post-treatment and maintained this level of tight repression throughout the 90-minute time course (Figure 2A). The addition of 10 nM of BASA resulted in a 14-fold repression of *bioD* expression 15 min post-addition (*p* ≤ 0.0001, Figure 2B). Similarly, this level of repression was maintained throughout the 90-minute time course. Stronger repression of *bioD* was achieved by increasing the dosage of BASA to 3.9 μM. A 52-fold decrease in *bioD* transcripts was measured at 15 min, and *bioD* expression was maintained over the time course with a 32-fold decrease measured at 30 min and 71-fold decrease at 90 min (*p* ≤ 0.0001, Figure 2C). The *bioY* transcript exhibited less dramatic repression than *bioD* for both supplemented biotin and BASA. With the addition of 10 nM biotin, *bioY* was only repressed two-fold at 15 min (*p* < 0.01), five-fold at 30 min (*p* < 0.01) and six-fold at 90 min (*p* < 0.01; Figure 2D). *S. aureus* treated with 10 nM of BASA exhibited a similar response, with *bioY* expression reduced by 1.5-fold at 15 min (*p* < 0.01), two-fold at 30 min (*p* < 0.01) and six-fold at 90 min (*p* ≤ 0.0001, Figure 2E). Together, these three lines of evidence demonstrated that BASA binds directly to the BPL target and functions as a co-repressor, promoting transcriptional repression comparable to that of the natural co-repressor biotinyl-5′-AMP.

### 2.3. Advanced Resistance Studies Identify Two Resistance Mechanisms

The ability of *S. aureus* to develop resistance to BASA was investigated using both spontaneous resistance and serial passage approaches. The *S. aureus* NCTC 8325 strain was selected due to the availability of a sequenced reference genome for comparative sequence analysis. To measure the rate at which spontaneous resistance develops, 10^9^
*S. aureus* cells were plated onto media containing four times the MIC of BASA. No resistant mutants grew after 24 hours which equates to a calculated resistance frequency of <10^−9^ at this inhibitor concentration. BASA resistant *S. aureus* strains were subsequently generated by the serial passage of seven populations of *S. aureus* NCTC 8325 in sub-inhibitory concentrations of BASA for 18 days at which time individual colonies were isolated. These were then passaged on non-selective media to verify that the inherited resistance was genetically stable. The sensitivity of the strains to BASA was then established using antimicrobial sensitivity assays (Table 2 and Appendix A). The isotypic parent strain was highly sensitive to BASA, with an MIC of 0.0625 µg/mL. Interestingly, the most resistant strain, B7, still retained a low MIC of 4 μg/mL (64-fold greater than non-treated control) suggesting complete resistance to this BASA is difficult to readily evolve. The MIC values for B1–B6 were a modest 2–16-fold increase over the parent strain.

To determine if there was a fitness cost associated with these mutations, in vitro growth curves for the seven strains were analyzed and doubling times calculated (Table 2, Appendix A) [37]. All strains had growth kinetics comparable to the parental strain NCTC8325 (52 min), except B3, which exhibited significantly inhibited growth (doubling time, 193 min, *p* < 0.0001 B3 vs. NCTC8325) preventing it from reaching half the maximum optical density of the other strains. Collectively, these data suggest that in vitro resistance to the BPL inhibitor is possible without significant associated fitness costs. To test if the resistance mechanisms identified were specific for BASA, all seven strains were subjected to antibacterial susceptibility assays against a panel of diverse antibiotics that included two known BPL inhibitors, namely biotinol-5′-AMP [32] and BPL223 (compound 4c in [30]). The evolved resistance mechanisms appeared specific to the BPL inhibitors as there was no change in susceptibility to β-lactams amoxicillin and methicillin (Appendix A). The only exception was observed for strain B3 which exhibited a modest two- to four-fold decrease in susceptibility to both β-lactams. Conversely, all strains exhibited decreased susceptibility to both BPL inhibitors. Whereas parent strain NCTC8325 exhibited an MIC of 1 μg/mL against biotinol-5′-AMP, all strains were resistant to this compound (MIC >64 μg/mL), except B5, which exhibited an MIC of 8 μg/mL. Similarly, NCTC8325 also exhibited an MIC of 8 μg/mL for BPL223, however B2, B4, B6 and B7 were not susceptible again (MIC >64 μg/mL). B1 (MIC = 64 μg/mL) and B3 (MIC = 16 μg/mL) exhibited a modest decrease in susceptibility, and B5 (MIC = 8 μg/mL) showed no change.

Whole-genome sequencing was subsequently employed to characterize mutations arising in the seven strains generated from advanced resistance studies, alongside a non-treated NCTC 8325 parent strain. DNA sequencing was performed with 150 bp paired-end reads on the Illumina Miseq platform, with each strain having 434,000–809,000 reads (between 22 and 40 times coverage of genome) sufficient for detection of single nucleotide variants [38]. The parent NCTC 8325 strain was compared to the reference sequence and differences observed between the parental strain and reference genome were accounted for in a new reference sequence. The resistant strains were then compared to this new parental reference genome. Between two and four mutations were identified in each strain (phenotypes summarized in Table 2, full characterization of genotypes in Appendix A). Collectively, this encompassed 15 non-synonymous mutations in the coding regions of 11 independent genes. The majority of these were single nucleotide variants. However, two episodes of insertions that mediated frame shifts in coding regions were observed as were five deletion events, ranging from three base pairs (i.e., a single codon) to larger deletions (194 to 200 bp). Four strains also contained 94 to 139 bp deletions between open reading frames, SAOUHSC_01476 and SAOUHSC_01477, approximately 1 kb upstream of the putative operon containing BPL encoding gene *birA* (ORF SAOUHSC_01473) which is predicted to span SAOUHSC_01476 to SAOUHSC_01471 [39].

The only gene with non-synonymous mutations in five independent strains was *pyc* which encodes the biotin-dependent enzyme PC. Strains B1, B2, B4, B6 and B7 all contained mutations that led to premature truncation of the PC protein. These strains also exhibited the greatest (4–64 fold) increases in MIC (Table 2). Streptavidin blot analysis performed on whole cell lysates to detect biotinylated protein confirmed the loss of biotinylated PC (Appendix A). In addition to the deletion of *pyc*, the most resistant strain B7 also contained the missense mutation D200E in the *birA* gene that encodes BPL. This was accompanied by a nonsense mutation in the *fmtA* gene, involved in removing D-alanine from teichoic acids in the cell wall [40], and an intergenic deletion between SAOUHSC_01476 and SAOUHSC_01477 (Appendix A). Other mutations present in strains containing the *pyc* deletion included the following: Strain B1 contained a frame-shift mutation, leading to premature truncation of the potassium transport protein *trkA* and a missense mutation, E20K, in a putative sensor histidine kinase. Strains B2, B4 and B6 contained deletions in the same intergenic region as B7, and strain B6 contained two additional missense mutations, P626L in the RNA polymerase β-subunit and R296H in a glycosyltransferase-A-like gene. It is worth noting that isolates from B6 and B7 were suspected to be derived from the same intermediate strain, possibly due to cross-contamination, as they contained identical mutations in two separate locations in the genome. Strains B3 and B5 that exhibited the lowest (two- to four-fold) reduced susceptibility to BASA were devoid of mutations in the *pyc* gene. Strain B3 contained two mutations; a missense mutation in the ci-di-AMP phosphodiesterase *gdpP* [41] that converted histidine 442 in the presumptive catalytic triad in the DHH domain to proline and a 3bp deletion which led to the loss of glutamine at position 213 in the disulfide stress effector *yjbH* [42]. Strain B5 contained a 111 bp deletion in a gene predicted as a *greA* transcription factor elongation homolog and a missense mutation P15L in a putative bacterial ferredoxin, SAOUHSC_01504. Interestingly, none of the strains harbored mutations in biotin-dependent ACC. Together these data suggest *S. aureus* can evolve at least two possible resistance mechanisms in vitro to overcome BASA: (1) deletion of *pyc* and (2) mutation of the BPL target.

### 2.4. Resistance Mechanism 1: Deletion of pyc

Of all the mutations identified in the advanced resistance study, four were predicted to cause a loss of function due to protein truncation (*pyc, trkA, fmtA, greA*), whilst two encoded for missense mutations (*gdpP, yjbH*). To determine if the loss of function in any of these gene products was sufficient for reduced susceptibility, strains containing transposon disrupted target genes were obtained from the Nebraska transposon mutant library [43]. As all mutants in the library were generated in *S. aureus* JE2 (CA-MRSA, USA300) this served as the isotypical control strain. Of the 10 discrete genes harboring mutations identified in B1–B7, five transposons disrupted strains were available for further analysis, namely *pyc, trkA, fmtA, gtfA* and *yjbH*. All five strains were initially tested for susceptibility to BASA by monitoring growth for 20 hours in the presence of varying concentrations of the compound. Parental strain JE2 exhibited an MIC of 0.5 μg/mL. Only transposon disruption of Δpyc and Δ*yjbH* lead to an increased MIC. Both strains exhibited an MIC of 4 µg/mL, the same susceptibility as that observed in the initial advanced resistance studies with isolate B7 (Table 2). To test if the deletions leading to reduced susceptibility also resulted in changed growth kinetics (as had been observed in strain B3 containing the mutation in *yjbH*), the growth kinetics were explored further for the Δpyc and Δ*yjbH* strains by monitoring growth over two days in the presence of 0, 0.5, 2 and 4 μg/mL BASA (Appendix A). The strains showed no difference in growth in the untreated control, suggesting that there was no growth defect associated with the mutations (Appendix A). Consistent with the MIC determination, the parental strain had no growth at concentrations higher than 0.5 μg/mL. Whilst growth of the Δ*pyc* and Δ*yjbH* strains was still partially inhibited by BASA, even at low concentrations, both strains showed significantly increased growth over the wildtype with 50% and 25% maximal growth relative to untreated control after two days (Appendix A). The Δ*pyc* strain also grew at 2 μg/mL (Appendix A) and Δ*yjbH* at both 2 and 4 μg/mL BASA (Appendix A). The growth kinetics of the strains under antibiotic pressure were also markedly different. The Δ*pyc* strain exhibited a pronounced lag phase before entering a relatively rapid log phase growth. In contrast, the Δ*yjbH* strain showed slower steady growth throughout the time course. The fact the Δ*pyc* strain was able to grow at four times the MIC suggested this mutation alone was sufficient for the four-fold increase in resistance observed in the NCTC-8325-derived strains containing this mutation.

### 2.5. Resistance Mechanism 2: Missense Mutation in the BPL Target

The advanced resistance studies also identified the missense mutation *Sa*BPL-D200E in clone B7. Analysis of the available structural data revealed that this mutation resides in the dimerization interface between the two BPL subunits (Appendix A). Based on this observation, we subsequently explored the proposal that *Sa*BPL-D200E disrupted dimerization activity. Site-directed mutagenesis was employed to re-engineer the D200E substitution into the protein sequence from *S. aureus* Mu50 for which structural data has been published [27,44]. Recombinant *Sa*BPL and *Sa*BPL-D200E were purified in their apo state for further characterization. Denatured mass spectrometry (MS) confirmed the desired mutation was present in *Sa*BPL-D200E (measured mass: 37,904 Da; predicted mass: 37,904 Da) and native nano-electrospray ionization–mass spectrometry (nESI–MS) together with a previously described streptavidin blot method both confirmed that the proteins were devoid of any co-purified biotinyl-5′-AMP (Figure 3 and Appendix A). The catalytic properties of wildtype and *Sa*BPL-D200E were then measured using an in vitro protein biotinylation assay. The biochemical data revealed that enzyme activity was not significantly affected by the D200E mutation, with the affinity for biotin increased modestly by two-fold (wildtype *Sa*BPL *K*_m_ 1.8 ± 0.3 μM, *Sa*BPL-D200E *K*_m_ 3.8 ± 0.4 μM (*p* < 0.05)). Likewise, the D200E mutation had a minimal effect upon the inhibitory activity of BASA in the enzyme assay (*K*_i_ wildtype *Sa*BPL 4.8 ± 2.1 nM; *K*_i_
*Sa*BPL-D200E = 10.9 ± 3.5 nM; *p* = 0.18). Native nESI–MS was then utilized to investigate the oligomeric state of *Sa*BPL-D200E (Figure 3, Appendix A and Appendix A). This technique has previously been employed to study co-repressor induced dimerization of wildtype *Sa*BPL [25]. Whilst *Sa*BPL underwent a well-characterized monomer-to-dimer transition induced by biotinyl-5′-AMP (Figure 3A,B), no dimerization was observed for *Sa*BPL-D200E in both its apo and holo forms across a range of concentrations (Figure 3C,D). This point mutation, thus, prevents protein dimerization. The native nESI–MS provided sufficient mass resolution to confirm the presence of bound biotinyl-5′-AMP, indicating that lack of protein dimerization is not due to the inability to catalyze the formation of co-repressor.

The effect of impaired homodimerization ability on DNA binding and transcriptional repression was then assessed using a combination of in vitro EMSA analysis and a previously established in vivo transcriptional reporter system in *E. coli* [23]. Both wildtype and *Sa*BPL-D200E demonstrated weak DNA binding activity in their apo states in the EMSA, with unbound *bioO* and *bioY* probes observed even at the highest protein concentration tested (i.e., 625 nM; Figure 4A–D). EMSA analysis was then repeated in the presence of biotin and MgATP to measure DNA binding in response to co-repressor biotinyl-5′-AMP. Unlike the apo protein, 50 nM of wildtype *Sa*BPL was sufficient to bind all of the oligonucleotide probes consistent with the homodimer having a higher affinity for the DNA (Figure 4E,G) [25,26]. Conversely, the D200E mutation had a significant impact on DNA binding. Approximately three-fold higher concentrations of holo *Sa*BPL-D200E were required for the same response with *bioO* (156 nM; Figure 4F) and the mutant failed to achieve complete binding to the *bioY* oligonucleotide even at the highest protein concentration tested (156 nM; Figure 4G,H). These data reveal that the D200E mutation impaired *Sa*BPL dimerization and reduced binding to DNA.

Finally, the biological consequence of the D200E mutation upon repressor activity was determined using an in vivo reporter assay. This in vivo reporter assay was previously developed in a biotin auxotrophic *E. coli* strain that had the DNA binding activity of its endogenous BPL removed [45]. The assay was adapted such that the transcriptional repressor activities of both wildtype *Sa*BPL and *Sa*BPL-D200E could be investigated against the *S. aureus bioO* and *bioY* promoters (Figure 5A). Expression of a β-galactosidase reporter gene was measured in the presence of varying biotin concentrations, and a repression constant determined (*K*_R biotin_ i.e., the biotin concentration required for a 50% reduction of β-galactosidase activity). A previously characterized *Sa*BPL-F123G mutant that similarly disrupts dimerization activity was also included for comparison [25,26]. Modest three-fold increases (*p* < 0.05) in biotin concentration were required for repression of the *bioO* promoter by either of the mutant proteins (D200E *K*_R biotin_ = 13.9 ± 3.4 nM; F123G *K*_R biotin_ = 15.3 ± 3.5 nM) compared to wildtype *Sa*BPL (*K*_R biotin_ = 4.3 ± 1.9 nM; Figure 5B). Greater differences were observed for the *bioY* promoter, where wildtype BPL (*K*_R biotin_ = 8.2 ± 0.7 nM) repressed at vastly lower biotin concentrations than either mutant (*K*_R biotin_ >500 nM), neither of which reached basal levels of expression even at 500 nM biotin (Figure 5C). These data were consistent with the reduced affinity of *Sa*BPL-D200E for the *bioY* oligonucleotide measured in the EMSA analysis. Together these data revealed that the dimerization-impaired mutant, *Sa*BPL D200E, reduced affinity for DNA binding in vitro and transcriptional repressor activity in vivo. The data also suggest impaired repression of the genes required for biotin synthesis as a resistance mechanism to combat BASA in *S. aureus*.

## 3. Discussion

BPL has been proposed as a drug target for the development of novel anti-infective therapeutics against antibiotic-resistant bacteria. In this report, we first investigated the mechanism of action of the potent BPL inhibitor BASA as an exemplar for this new class of antibacterial. Multiple lines of evidence were consistent with BASA acting directly through the BPL target. It has previously been proposed that anti-BPL compounds exert their antibacterial activity through inhibition of the enzymatic function of BPL and consequent reduced biotinylation and activity of the two biotin-dependent enzymes present in *S. aureus*, namely PC and ACC [7]. The data presented here provide evidence that anti-BPL compounds can also act as co-repressors to repress biotin synthesis and transport. In support, transcriptomic studies revealed that BASA induced repression of both *bioY* (biotin transport) and *bioD* (biotin synthesis) in *S. aureus* with similar potency and kinetics as the natural ligand biotin. These data are consistent with the hypothesis that the pharmacological inhibition of BPL induces biotin starvation within the bacteria by compromising both the supply and utilization of biotin. The combined effect of these mechanisms contributes to the antibacterial efficacy of BASA. Biotin deprivation is detrimental to the bacteria as neither PC nor ACC can be post-translationally activated by BPL without bioavailable biotin. In response, the bacteria attempt to increase biotin supply through either *de novo* synthesis (using *bioO*) or import of exogenous sources (using *bioY*). The importance of the biotin starvation state in the activity of anti-BPL compounds has not been previously reported.

Having established a greater understanding of the mechanism of action of BASA, we next addressed possible mechanisms of resistance. The incidence rate for the generation of spontaneous resistance was low, with no resistant colonies growing after 24 hours, resulting in a resistance rate of less than 1 × 10^−9^. Similar low spontaneous resistance rates have been reported for another anti-BPL compound active against *M. tuberculosis* (less than 1.4 × 10^−10^) [33]. This highlights BPL as an important antibacterial target with low potential for resistance to develop. Our advanced resistance studies involving serial passaging of *S. aureus* in sub-inhibitory concentrations of BASA for 18 days and subsequent characterization of the resistant strains resulted in the identification of two discrete resistance mechanisms; (1) deletion of PC and (2) a missense mutation in the BPL target. Loss of PC was the most common resistance mechanism observed in laboratory testing. A premature truncation of PC was present in five of the seven strains evolved to be resistant to BASA. As the biotin-accepting domain of PC that is essential for catalysis is contained within the C-terminal region of the enzyme, these truncated proteins were unable to be biotinylated and, hence, inactive. The transposon disrupted Δ*pyc* strain in the genetic background of MRSA strain USA300 JE2 confirmed that PC deletion conferred resistance independently of the other mutations and was a viable resistance mechanism in another genetic background, exhibiting a four-fold decrease in MIC over the isotypic wildtype parent strain. Notable was the observation that no mutations were observed in ACC. The current report suggests that bioavailable biotin is selectively utilized such that ACC is preferentially biotinylated over PC. This was consistent with previous genetic knockout studies which have demonstrated ACC, but not PC, is essential for growth in laboratory cultures [9,10]. Removal of PC from the cell, therefore, overcomes biotin starvation by removing the only other competing protein that requires biotin. However, this is unlikely to be a mechanism readily available to *S. aureus* during infection in vivo, as loss of PC and other tricarboxylic acid cycle genes have been shown to reduce virulence in murine and nematode models of infection [20,21].

The advanced resistance studies also identified a strain with a missense mutation in BPL. Biochemical characterization of this mutation revealed disrupted homodimerization, consequently leading to weakened DNA binding and reduced repressor activities. This residue, D200, is located on a loop within the dimer interface that becomes ordered upon ligand binding. The side chain of Asp200 forms hydrogen bonding interactions with Arg122 and hydrophobic interactions with Phe123 on the opposing subunit. By inhibiting dimer formation, D200E disrupted the DNA binding activity of *Sa*BPL and repression of *Sa*BPL target genes. The mechanism of resistance findings supports the proposal that the co-repressor function of BASA contributes to antibacterial efficacy. We propose that disruption of the repressor activity provides an opportunity for cells to resist the action of BASA by increasing the availability of intracellular biotin. The D200E mutant with reduced repression of *bioO* and *bioY* will result in increased *de novo* biotin synthesis and biotin transport, respectively. Increased accumulation of intracellular biotin could compete against BASA for binding within the active site of *Sa*BPL. In support, we observed that increasing exogenous biotin concentrations decreased the antibacterial activity of BASA. It has also been shown that *S. aureus* can stimulate import and sequester biotin inside the cell in response to increased exogenous biotin concentrations [25]. Similar resistance mechanisms through the deregulation of target genes have also been observed for other antibiotic targets in the literature. The antibiotic pyrithiamine binds a riboswitch protein and causes the repression of thiamine biosynthesis genes, leading to thiamine starvation. However, resistance to pyrithiamine is generated through mutations in the binding site of the riboswitch protein such that the antibiotic can no longer bind and cause gene repression, leading to a de-repression of thiamine synthesis [46,47].

The mechanism of entry of anti-BPL inhibitors into *S. aureus* has not been established. One possibility is that their common biotinyl moiety facilitates import into the bacteria through BioY, the biotin-specific subunit of the multivitamin transport system. Hence, targeting BPL with an inhibitor that also acts as a co-repressor would reduce BioY expression and limit entry of the chemotherapeutic into the bacteria which would be an undesirable situation for an antibacterial. However, several lines of unpublished data suggest that the BASA enters the bacteria through an alternative mechanism, likely via membrane-associated aquaporins. No change in the antibacterial potency of BASA was observed when tested against a *bioY* knockout *S. aureus* strain (data not shown). Similarly, increasing the concentration of exogenous biotin in the growth media did not impede the internalization of a fluorescent BPL inhibitor containing the biotinyl group [30]. Together these observations suggest that BASA is imported into bacteria independently of BioY. Hence, the altered expression of BioY noted here is unlikely to impact the accumulation of BASA inside the bacteria and affect its efficacy.

The oligomeric state of the *Sa*BPL D200E variant reported here supports previous in vitro studies on both *S. aureus* and *E. coli* BPLs, where modification of residues in the dimerization interface affect repressor activity. The mutations, F123G in *Sa*BPL [25,26] and D197Y in *E. coli* (equivalent of D200 in *S. aureus*) [48,49] demonstrate impaired homodimerization and loss of repressor activity [25,48,49]. However, it should be noted that the mutants used in those studies were engineered using site-directed mutagenesis based on knowledge of the crystal structure. The current report provides the first example where this phenotype has been evolved in vivo. The interaction between Phe123 and Asp200 appears particularly important in *Sa*BPL dimerization. Asp200 from one subunit forms hydrophobic interactions with Phe123 on the opposing subunit to facilitate dimerization. Mutation of these two amino acids result in enzymes with similar affinity for biotin, and only two-fold weaker than the wildtype enzyme [26]. Therefore, the ligand binding and catalytic properties of the mutant were not impaired. Conversely, homodimerization was significantly disrupted for the mutants, thereby perturbing DNA binding and repressor activities. It is proposed that these mutant strains require a much higher concentration of biotin before *Sa*BPL can be induced to be a fully active transcriptional repressor. Additionally, crystal structures of *Sa*BPL in complex with BASA reveal that the inhibitor becomes buried deep in the homodimer and is shielded away from the solvent. Consequently, it is difficult for the inhibitor to readily diffuse out from the binding site when *Sa*BPL is in its dimeric state. This results in a stable inhibitor-protein complex that ensures the compound experiences high residency time with the BPL target, thus contributing to the overall efficacy of the antibacterial.

## 4. Materials and Methods

### 4.1. General Bacterial Culture and Molecular Biology Reagents

All chemicals and reagents were purchased as analytical grade or higher. All molecular biology enzymes (restriction enzymes and DNA polymerase) and buffers were acquired from New England Biolabs. Oligonucleotides were purchased from Geneworks Ptd Ltd. Details of oligonucleotides used for cloning (Appendix A), RT–PCR (Appendix A) and EMSA analysis (Appendix A), bacterial strains (Appendix A) and plasmids (Appendix A) used in the study are supplied in the Appendix A. Unless otherwise stated, all bacterial strains were purchased from the American Tissue Culture Collection. The following strains were provided by the Network on Antimicrobial Resistance in *Staphylococcus aureus* (NARSA) for distribution by BEI Resources (BEIResources.org), NIAID, NIH: *Staphylococcus aureus* subsp. aureus, Strain USA300 (JE2), Transposon Mutants NE754, NE896, NE1022, NE788 and NE792. Cation-adjusted Mueller Hinton II (Becton, Dickinson and Company, MD, USA) was used to propagate *S. aureus*. Antibacterial agents were obtained from Sigma-Aldrich.

### 4.2. Quantification of Gene Expression Using qRT–PCR

The qRT–PCR protocols used to measure the expression of *bioD* and *bioY* were completed as outlined previously [23]. However, the sub-cultured *S. aureus* NCTC 8325 cultures were treated with either 10 nM biotin or BASA at 10 nM or 3.9 μM (i.e., 4× MIC) once reaching mid-log phase, prior to cell lysis, RNA extraction and qRT–PCR analysis.

### 4.3. Electrophoretic Mobility Shift Assay (EMSA)

The EMSA protocol was adapted from methods previously published [23,25,50]. For analysis of BASA, 100 μM of the compound was added to the EMSA binding buffer. As BASA was dissolved in 100% DMSO, the final DMSO concentration in the binding buffer was adjusted to 2.5% (*v/v*). Control reactions were prepared by adding 100 μM biotin and 2.5% (*v/v*) DMSO to the binding reaction.

### 4.4. Spontaneous Resistance Rate

A culture containing 10^9^ CFU of *S. aureus* NCTC 8325 was plated onto cation-adjusted Mueller Hinton broth (CAMHB) Agar containing 4× MIC of BASA. Nine technical replicate cultures were plated from the same starting colony to control for priming mutations in the initial colony. Plates were assessed for growth after 24 h at 37 °C. As the determination of true resistance rates using fluctuation analysis requires some colonies to grow [51,52,53], a resistance frequency was instead calculated.

### 4.5. Generation of Resistant Mutants by Serial Passaging

Mutants were selected by serial passage in CAMHB in 96-well plates as previously described [54]. Briefly, a 2-fold dilution series of BASA was plated across 7 wells. Initial concentrations ranged from 1 μg/mL to 0.016 μg/mL, with these concentrations increasing (up to 64 μg/mL) as the MIC increased. Approximately 10^4^ CFU of *S. aureus* NCTC 8325 at mid-log phase were added to each well and allowed to grow at 37 °C for 20 h. The OD at 620 nm was measured and the highest concentration of BASA that permitted growth (OD_620_ > 0.1 used as the threshold) was diluted 1000-fold to inoculate a new concentration series of BASA. This process was repeated for 18 days, and on the final day, single isolates were recovered by streaking out the strains from the highest concentrations of BASA that still allowed growth. Genomic DNA was isolated with the Wizard Genomic DNA purification kit (Promega, Wisconsin, USA) according to the manufacturer’s guidelines. Bacteria were treated with 10 μg of lysostaphin (Sigma-Aldrich Inc, MO, USA) and 1 mg/mL lysozyme prior to lysis. These selected strains were also propagated for 6 days on CAMHB agar in the absence of antibiotic selection, before re-testing these strains for antibacterial susceptibility to assess the stability of the generated resistance.

### 4.6. Genomic DNA Purification and Whole-Genome Sequencing

Genomic DNA was isolated with the Wizard Genomic DNA purification kit (Promega, Wisconsin, USA) according to the manufacturers’ guidelines. Bacteria were treated with 10 μg of lysostaphin (Sigma-Aldrich Inc, MO, USA) and 1 mg/mL lysozyme prior to lysis. DNA quantities ranged from 15–60 ng/μL. Genomic DNA from parental strain NCTC 8325 and isolates B1–B7 was submitted to ACRF (Cancer Genomics Facility, Adelaide, South Australia) for library preparation and sequencing. Whole-genome sequencing was performed with 150 bp paired-end reads on Illumina Miseq. The number of sequences for each strain ranged from 434,000–809,000 (between 22 and 40× coverage of genome), 32% GC content. Sequence reads are stored on the Biosample with accession numbers B1: SAMN14388873, B2: SAMN14388874, B3: SAMN14388875, B4: SAMN14388876, B5: SAMN14388877, B6: SAMN14388878, B7: SAMN14388879, *S. aureus* subsp. *aureus* NCTC 8325 SAMN14388880.

### 4.7. Bioinformatic Analysis

The Fastq reads were analyzed using the Breseq analysis pipeline software [55]. The parental NCTC 8325 strain that had not undergone antibacterial selection was mapped to the reference genome (NC_007795.1) and the differences were applied to generate a new reference sequence using gdtools. The de-multiplexed reads from isolates B1–B7 were then mapped to this new reference and mutations identified. More than 98% of reads from each isolate were mapped to the reference genome. Manual inspection of the resulting HTML files was used to determine true positives. Amongst those manually excluded from further analysis was a missense mutation in *rsbU* which is not transcribed in NCTC8325 [56], mislabeling of P626L (as P579L) due to a non-standard start codon and removal of mutations identified in all strains (including the parental NCTC 8325) that did not reach the threshold required for inclusion into the new reference genome.

### 4.8. Antimicrobial Susceptibility Testing/Cross Resistance

Antimicrobial susceptibility assays were performed using a broth microdilution method as recommended by CLSI (Clinical and Laboratory Standards Institute, Document M07-A8, 2009, Wayne, Pa.). Antibacterial compounds were diluted two-fold in cation-adjusted Mueller Hinton Broth (CAMHB) to a final concentration range of 64–0.06 μg/mL in 3.2% DMSO. To prepare the inoculum, an overnight culture of *S. aureus* was subcultured 1:1000 into fresh media and grown to mid-log phase. Then, 96-well microtiter plates containing diluted compounds were inoculated with 5 × 10^4^ CFU and incubated at 37 °C for 20–22 h with shaking. Growth of the culture was quantified after shaking for 15 seconds to disrupt any sedimentation by measuring the absorbance at 620 nm or 630 nm using either a Thermo Multiskan Ascent plate reader or Biotek EL808 plate reader, respectively. Assays were performed in triplicate.

### 4.9. Measurement of the Kinetics of Bacterial Growth

The kinetics of bacterial growth were measured in Mueller Hinton broth devoid of antibiotics. Growth curves were obtained by subculturing an overnight culture 1:1000 and incubating cells at 37 °C to an approximate OD_600_ of 1 before a further 1:1000 dilution to yield an approximate initial inoculum of 10^5^ cells in 100 μL. Cultures were grown for 24 h in a 96 well plate at 37 °C. Growth of the cultures was monitored at OD_630_ every 30 min for 24 h with shaking using an EL808™ Absorbance Microplate Reader (BioTek Instruments Inc, Winooski, VT, USA). Growth curves were analyzed by the method described in [37]. In brief, the log of the ODs from OD 0.1–0.3 was plotted against time and log(2)/gradient was determined giving the doubling time. Cultures were plated in technical duplicate and three independent biological replicates were collected.

### 4.10. Protein Methods

Expression and purification of non-liganded (i.e., apo) *Sa*BPL were performed following previously described protocols [25,44,57]. Purification of apo protein was performed by incubating bacterial lysate with streptavidin agarose at 30 °C for 1 h prior to incubating with purified biotin domain-GST fusion protein from *S. aureus* [58] (*Sa*PC90) at 30 °C for 1 h, as described previously [25,26]. Protein concentration was determined by Bradford assay. Purified apo protein was confirmed by native nano-electrospray ionization–mass spectrometry (nESI–MS) and by Western blot analysis as described previously [25,26]. An enzyme activity assay to measure the incorporation of biotin onto protein by BPL was performed as previously reported [30].

### 4.11. Mass Spectrometry

Native nESI–MS of apo *Sa*BPL was carried out on a Waters Synapt G1 HDMS system completed according to [25]. MS parameters were as previously reported to maintain non-covalent interactions and included; capillary voltage, 1.5–1.7 kV; cone voltage, 40–80 V; trap collision energy, 20–50 V; transfer collision energy, 15–20 V; source temperature, 50 °C; extraction cone, 2.0–5.0 V; trap gas flow, 5–8 mL/min; backing pressure, 3.95 mbar. Data analysis was performed in MassLynx V4.1 using manual peak finding.

### 4.12. Chromosomal Integration and β-Galactosidase Reporter Assay

*Sa*BPL-D200E was cloned into the established integration vector (methods provided in Appendix A) and incorporated chromosomally into the *E. coli* reporter strains prepared previously [23,25], using established methods [45]. β-galactosidase assays were performed as previously reported [23,25,45].

## 5. Conclusions

In this report, we demonstrate that BASA exerts its antibacterial effect through two mechanisms inhibiting the catalytic biotinylation activity of BPL and serving as a co-repressor. BASA treatment results in a biotin starved state within the bacteria. *S. aureus* strains resistant to BASA were characterized and two discrete resistance mechanisms were discovered. However, the low spontaneous resistance rate would suggest that resistance may be difficult to evolve in vivo and confirms BPL as a bona fide antibacterial drug target. Further chemical development of BPL inhibitors will focus on maintaining both mechanisms of action. New analogous with improved pharmacology are under development to realize valid pre-clinical antibiotic candidates.

## Figures and Tables

**Figure 1 antibiotics-09-00165-f001:**
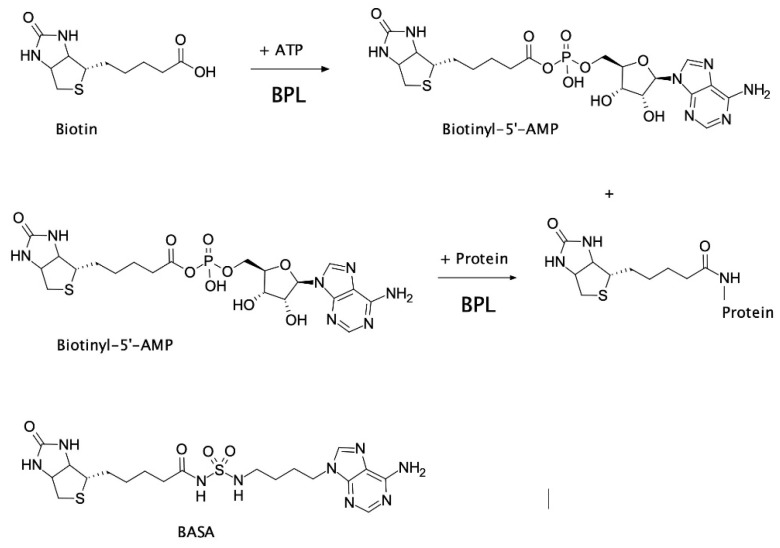
Protein biotinylation catalyzed by biotin protein ligase (BPL). Biotin ligates with ATP to produce the reaction intermediate, biotinyl-5′-AMP. Biotinyl-5′-AMP is employed to covalently attach the biotinyl moiety onto a specific lysine residue present in the biotin-accepting protein substrate. The chemical structure of biotinyl-acylsulfamide adenosine (BASA), a chemical analog of the reaction intermediate, is shown.

**Figure 2 antibiotics-09-00165-f002:**
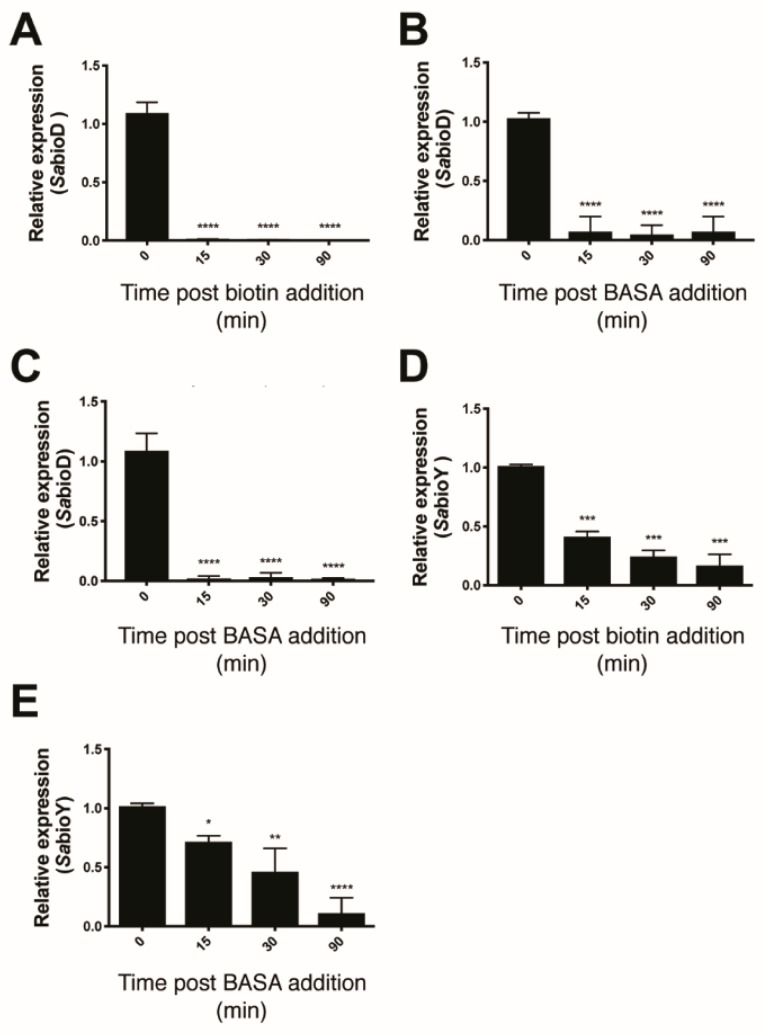
BASA represses expression of *bioD* and *bioY*. Quantitative RT–PCR was employed to measure the repression of *bioD* following treatment with (**A**) 10 nM biotin, (**B**) 10 nM of BASA (0.01× minimal inhibitory concentration (MIC)) and (**C**) 3.9 μM (4× MIC) of BASA, and the repression of *bioY* in response to (**D**) 10 nM biotin and (**E**) 10 nM of BASA. Transcript levels were normalized against an internal control (16s rRNA) and calculated relative to time = 0. Error bars represent SEM from 3 independent biological replicates (*n* = 3), **p* < 0.05, ***p* < 0.01, ****p* > 0.001, *****p* < 0.0001.

**Figure 3 antibiotics-09-00165-f003:**
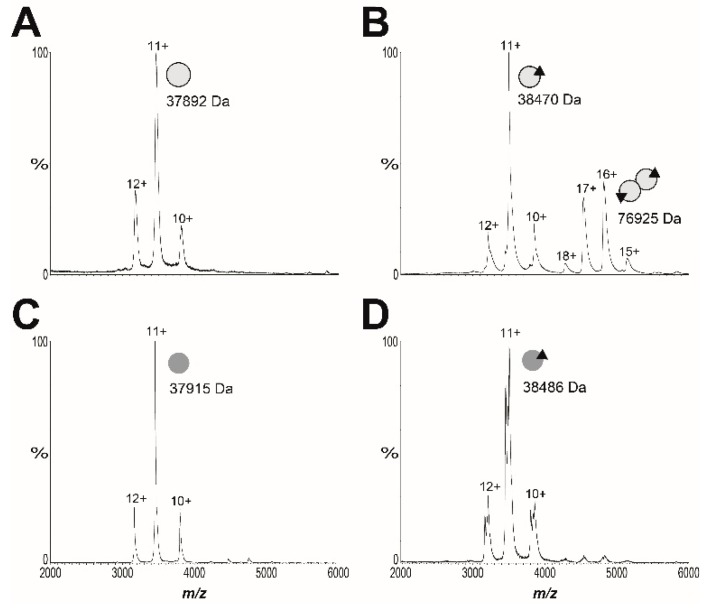
Native native nano-electrospray ionization–mass spectrometry (nESI–MS) reveals *S. aureus* BPL (*Sa*BPL) D200E is monomeric in the presence and absence of ligands. The oligomeric states of 10 μM of (**A**) wildtype apo *Sa*BPL and (**B**) holo *Sa*BPL were compared with 90 μM of (**C**) apo *Sa*BPL-D200E and (**D**) holo *Sa*BPLD200E. Monomeric protein is diagrammatically represented by a single grey sphere, homodimer by conjoined spheres and the presence of biotinyl-5′-AMP by a black triangle. Spectra showing *Sa*BPL D200E at varying protein concentrations between 1.4 and 90 μM are presented in Appendix A and masses extracted from the spectra are in Appendix A.

**Figure 4 antibiotics-09-00165-f004:**
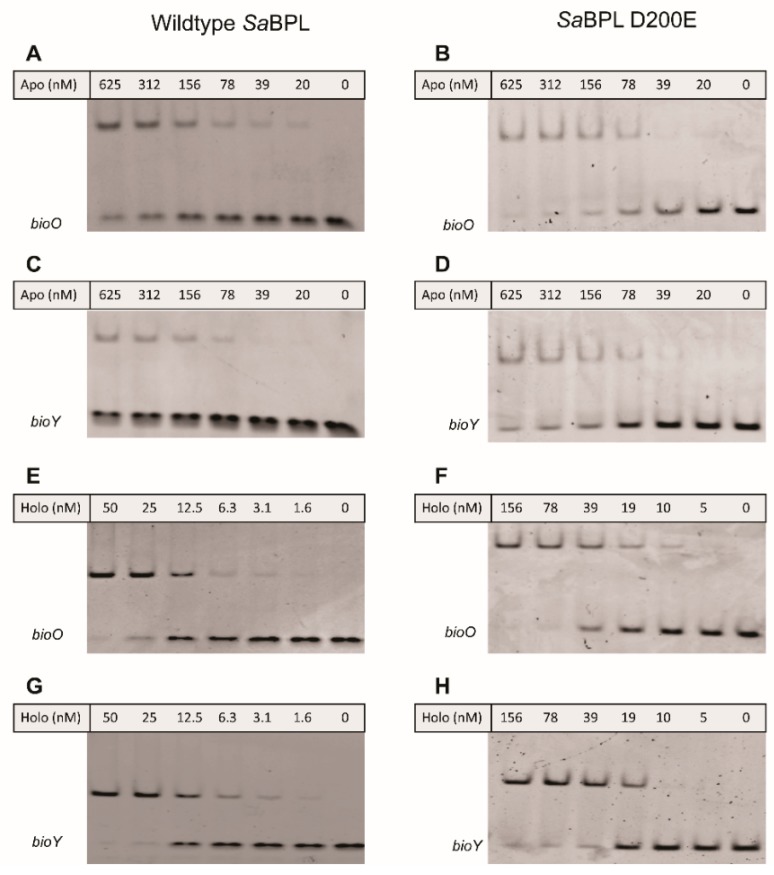
*Sa*BPL D200E has reduced DNA binding activity compared to wildtype *Sa*BPL**.** Electrophoretic mobility shift assays (EMSA) analysis was performed to measure the binding of (**A**) apo *Sa*BPL wildtype binding to *bioO*, (**B**) apo *Sa*BPL D200E binding to *bioO*, (**C**) apo *Sa*BPL wildtype binding to *bioY* and (**D**) apo *Sa*BPL D200E binding to *bioY*, (**E**) holo *Sa*BPL wildtype binding to *bioO*, (**F**) holo *Sa*BPL D200E binding to *bioO*, (**G**) holo *Sa*BPL wildtype binding to *bioY* and (**H**) holo *Sa*BPL D200E binding to *bioY*.

**Figure 5 antibiotics-09-00165-f005:**
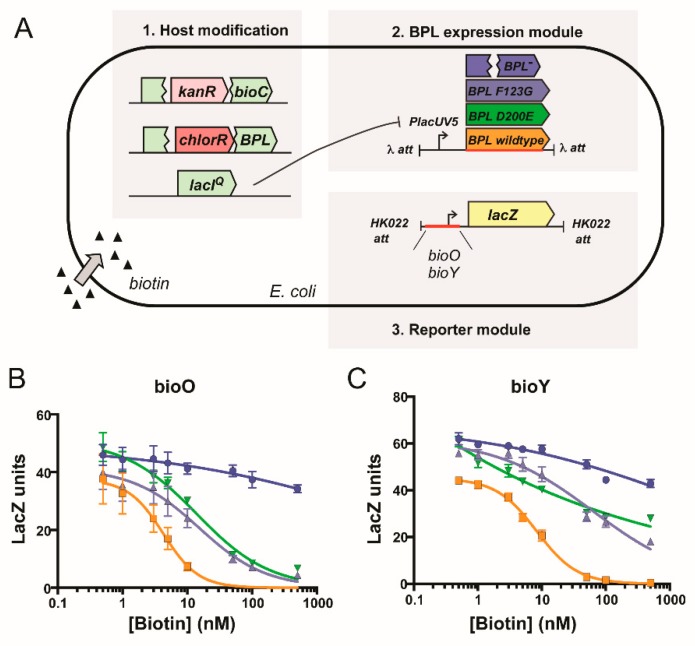
Biotin-regulated repressor activity of *Sa*BPL and mutants. (**A**) In vivo expression assays were performed in an *E. coli* reporter strain containing chromosomally integrated repressor and promoter constructs [23]. *LacZ* units were calculated by subtracting the value generated by the control strain containing no integrated promoter construct (*LacZ* unit ≤10). Expression of the β-galactosidse reporter gene, under the control of promoters (**B**) *bioO* or (**C**) *bioY*, was measured in media containing varying concentrations of biotin. Transcription factors *Sa*BPL (orange), *Sa*BPL D200E (green) and *Sa*BPL F123G (black) were analyzed alongside a control strain that harbored no repressor protein (blue). Error bars represent SEM from independent biological replicates (*n* = 6). The amount of biotin to reach half-maximum repression (*K*_R biotin_) was calculated from the curves using GraphPad Prism. The *K*_R biotin_ for wildtype *Sa*BPL repressing *bioO* and *bioY* was 4.3 ± 1.9 nM and 8.2 ± 0.7 nM, respectively. The *K*_R biotin_ for *Sa*BPL-D200E repressing *bioO* and *bioY* was 13.9 ± 3.4 nM and >500 nM, respectively. The *K*_R biotin_ for *Sa*BPL-F123G repressing *bioO* and *bioY* was 15.3 ± 3.5 nM and >500 nM.

**Table 1 antibiotics-09-00165-t001:** Antibacterial susceptibility and cytotoxicity of BASA.

Species	MIC (μg/mL)
*Staphylococcus aureus*	
Methicillin-sensitive (*n* = 8)	0.25–0.5
Methicillin-resistant (*n* = 9)	0.25–0.5
Coagulase negative Staphylococci (*n* = 7)	0.125–0.5
*S. aureus* ATCC 49775	0.5
*S. aureus* ATCC 49975 MBH + 10% FCS	0.5
*S. aureus* ATCC 49975 MHB + 20% FCS	4
*Mycobacterium tuberculosis* (*n* = 1)	55
*Streptococcus pneumoniae* (*n* = 6)	>32
*Enterocococcus faecalis* (*n* = 3)	>128
*Enterococcus faecium* (*n* = 5)	>128
*Escherichia coli* (*n* = 1)	>128
Cell lines	EC50 (μg/mL)
HepG2	>250
HEK293	>250

**Table 2 antibiotics-09-00165-t002:** Strains used in this study and relevant genetic and phenotypic changes associated with BASA resistance. MICs were determined against BASA. Significance for doubling times was determined by ANOVA analysis using Dunnett’s multiple comparison test. (****p* < 0.001). A full genotypic analysis of strains B1–B7 is shown in Appendix A.

Strain	MIC μg/mL BASA	Doubling Time (Min)	Genotype
NCTC 8325	0.065	51.8 ± 2.1	
B1	1	69.2 ± 11.6	Δ*pyc* *trkA* *pyc* Δ194bp (2761-2954/3453) SAOUHSC_01981 E20K
B2	0.5	64.5 ± 1.42	Δ*pyc* Δ94bp intergenic (SAOUHSC_01476 to 01477) presumed silent mutation (no *rsbU*)
B3	0.125	193.6 ± 14.3 ***	*gdpP, yjbH* *yjbH* Q213-
B4	0.25	53.5 ± 5.5	Δ*pyc* Δ113bp intergenic (SAOUHSC_01476 to 01477)
B5	0.25	68.8 ± 12.1	*greA* SAOUHSC _01504 P15L
B6	1	41.9 ± 0.73	Δ*pyc, rpo**β, gtfA* homolog Δ139bp intergenic (SAOUHSC_01476 to 01477) *rpoβ* P626L SAOUHSC_02984 (*gtfA-*like) R296H
B7	4	67.1 ± 0.90	Δ*pyc,* *birA* D200E, *fmtA* *pyc* Δ200bp (917-1116/3453) Δ139bp intergenic (SAOUHSC_01476 to 01477) *fmtA* K163
JE2	0.5	ND	Parent strain, USA300 JE2
NR-47297	4	ND	JE2 *Δpyc*
NR-47439	4	ND	JE2 *Δ**yjbH*
NR-47565	0.5	ND	JE2 *Δ**fmtA*
NR-47335	0.5	ND	JE2 *Δ**gtfA*
NR-47331	0.5	ND	JE2 *ΔtrkA*

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
