# Peer review of "Advanced Resistance Studies Identify Two Discrete Mechanisms in Staphylococcus aureus to Overcome Antibacterial Compounds that Target Biotin Protein Ligase"

_antibiotics, 2020, doi:10.3390/antibiotics9040165_

Round 1

Reviewer 1 Report

This interesting manuscript by Hayes et al. outlines a systematic study into the mechanisms of action and resistance of a novel anti-Staphylococcal molecule, BASA. They describe the dual activity of this molecule which inhibits biotin protein ligase and inhibits transcription of biotin-associated genes. They found that resistance develops infrequently, and characterised resistance that developed to two pathways: deletion/inactivation of pyruvate carboxylase, allowing available biotin to be used by Acetyl-CoA carboxylase, and secondly via a mutation in BPL itself.

This manuscript is very well written and clearly represents a substantial body of work. The authors have developed and tested their hypothesis using appropriate means. This work is timely, as new and specific antibiotics are desperately needed. The specificity of these inhibitors is ideal, as they could target the pathogen while leaving the microbiome largely intact. I have thoroughly enjoyed reading this manuscript.

I have a few minor comments and points for clarity.

  • The authors mention that the eukaryotic BPL would not be impacted by BASA, and their toxicity studies support this. Likewise, BASA is specific to S. aureus with some activity on Mycobacterium. I was curious as to how similar/divergent S. aureus BPL is from other bacteria (Gram positives and negatives), and from eukaryotic versions? Perhaps a brief sentence or two could be added to the introduction?
  • Supplemental tables are not numbered correctly with the order of the text, please could this be rectified?
  • In table S8 the strain JD26186 is used, could the authors provide a little more detail on what this strain is in text/table?
  • In lines 152-155: Could the authors please clarify if treatment occurs only after cultures reach mid-log phase?
  • I was pleased to see that after generating mutants the authors grew the lineages without selection to ensure mutations were stable.
  • In Figure 2A – were the transcripts measured in untreated controls at each time point? My concern is expression of some genes is growth phase dependent, and the authors are comparing their treated gene expression from different time points. I appreciate the BASA at 4xMIC will inhibit growth (as authors show it is bacteriostatic), but would the biotin treatment? Please could the authors clarify their choices in the text.
  • In lines 308 the authors state that BASA and biotin are “comparable” in transcriptional repression. It looks to me from Figure 2 that biotin is still much more potent than BASA – although both are very effective.
  • For convenience, in Figure 2 please could the authors indicate the relation of 10nM to MIC, as they do for 3.9uM?
  • For table 2, I expect the doubling times were performed in media with no antibiotics or inhibitors, but this is not clearly stated in the methods. Please could the authors add this?
  • In Figure S2, do the two different concentrations of paracetamol relate to the two different cell types? (please clarify in legend).
  • In Figure S6 – what is the solubility limit of biotin? I’m concerned that the higher concentrations start at a much higher OD than the lower concentrations. Please can the authors clarify why this is?

Author Response

1. The authors mention that the eukaryotic BPL would not be impacted by BASA, and their toxicity studies support this. Likewise, BASA is specific to S. aureus with some activity on Mycobacterium. I was curious as to how similar/divergent S. aureus BPL is from other bacteria (Gram positives and negatives), and from eukaryotic versions?

Response: We thank the reviewer for their interesting and thoughtful question. We have previously described the structural differences between BPLs in our comprehensive 2017 review in TIBS (Ref 8). The BPLs fall into three classes; Class I are a are simple module (eg mycobacteria), Class II are the bifunctional proteins, such as SaBPL described in the current paper, and Class III are the eukaryotic homologs. All BPLs share a conserved catalytic domain in the C-terminal regions where biotin and ATP binding (and BASA also). Class III BPLs do not have the DNA binding fold on their N-terminus like Class II enzymes, instead possess multi-domain an extension that is proposed to have a proof-reading activity that ensures only the correct protein substrates are selected for biotinylation. We propose that it the differences in the N-terminal extensions that possibly contribute to selective BASA binding, with the Class III enzymes possibly being able to exclude binding of substrates, such as BASA. This is still speculative, and further work is required to delineate these ideas. Accordingly, we have decide not to amend the introductions with speculative ideas.      

2. Supplemental tables are not numbered correctly with the order of the text, please could this be rectified?

Response: We have carefully checked both the manuscript and Supplementary material. We respectfully disagree with the reviewer – the supporting tables do indeed agree with the order they are presented in the text. To enhance the clarity, we have reworded the Materials section to better describe the contents of Tables S4 – S9.

3. In table S8 the strain JD26186 is used, could the authors provide a little more detail on what this strain is in text/table?

Response: E. coli strain JD26186 (bioC::Kan), based on host strain KP7600, is a transposon-disrupted bioC biotin auxotroph, and was obtained from the National BioResource Project (NIG, Japan). This has been previously described in Supp Reference 6.

4. In lines 152-155: Could the authors please clarify if treatment occurs only after cultures reach mid-log phase?

Response: I assume that the reviewer is asking about the antimicrobial susceptibility assays described in lines 152-155. The assay is performed following CSLI guidelines. Bacteria are indeed grown to mid log phase then diluted out prior to plating.

5. I was pleased to see that after generating mutants the authors grew the lineages without selection to ensure mutations were stable.

Response: Thank you for this endorsement. We believed this was an important step to include in the methodology.

6. In Figure 2A – were the transcripts measured in untreated controls at each time point? My concern is expression of some genes is growth phase dependent, and the authors are comparing their treated gene expression from different time points. I appreciate the BASA at 4xMIC will inhibit growth (as authors show it is bacteriostatic), but would the biotin treatment? Please could the authors clarify their choices in the text.

Response: We have performed the necessary controls described by the reviewer and reported the careful characterization of our assay in References 23 and 25. We have shown that the levels of these two target transcripts is not altered in untreated cells grown in biotin-depleted media and their expression remains upregulated throughout the 90 minute time course. Furthermore, we have reported that transcriptional repression is specifically induced by biotin (or biotin-like analogue such as BASA).

7. In lines 308 the authors state that BASA and biotin are “comparable” in transcriptional repression. It looks to me from Figure 2 that biotin is still much more potent than BASA – although both are very effective.

Response: We are in agreement with the reviewer that both biotin and BASA are both effective transcriptional co-repressors. Whilst biotin is slightly more potent, BASA treatment does significantly reduce the amount of target transcripts with similar kinetics as biotin.

8. For convenience, in Figure 2 please could the authors indicate the relation of 10nM to MIC, as they do for 3.9uM?

Response: We have included an extra sentence in the Figure legend to state that 10 nM is 0.01X MIC.

9. For table 2, I expect the doubling times were performed in media with no antibiotics or inhibitors, but this is not clearly stated in the methods. Please could the authors add this?

Response: We have added the additional sentence into the methods: The kinetics of bacterial growth were measured in Mueller Hinton broth devoid of antibiotics.

10. In Figure S2, do the two different concentrations of paracetamol relate to the two different cell types? (please clarify in legend).

Response: We have discovered that paracetamol does have slightly different ED50 values or the two different cell lines, and these concentrations are shown in the legend. No further changes are necessary.

11. In Figure S6 – what is the solubility limit of biotin? I’m concerned that the higher concentrations start at a much higher OD than the lower concentrations. Please can the authors clarify why this is?

Response: Biotin is highly soluble in growth media, in excess of 1 mg/ml (4.1 uM), so there were no issues with solubility in this assay. We have previously reported that biotin can promote the growth of S. aureus, and this explains the higher ODs.

Reviewer 2 Report

In this work a new antibiotic that acts on Biotin protein ligase (BPL) is studied. Its action and possible resistance mechanisms are described.

The following observations are made:

Line 37: What does BASA mean?

Line 74: What does TCA mean?

Author Response

1. Line 37: What does BASA mean?

Response: The full chemical name for BASA, biotinyl-acylsulfamide adenosine, appears on lines 124-5 in the Introduction where it is first introduced. It is not necessary to include this information in the abstract.

2. Line 74: What does TCA mean?

Response: TCA refers to the tricarboxylic acid cycle, also known as citric acid cycle or Krebs cycle. We have removed the TCA abbreviation and spelt this out in full in the revised draft.

Reviewer 3 Report

The authors characterize the mechanisms by which S. aureus develops resistance to BASA, a biotin analogue that both inhibits biotin dependent protein ligase (BPL) and expression of genes involved in the production of biotin.

This is a lengthy, detailed paper with many authors and aspects that include genetics studies, mass spectrometry, EMSA assays of protein-DNA interactions, mutagenesis and genome sequencing.

That said, it is well organized, well-documented, clear and cohesive in the arguments. Two small notes are offered:

  1. The authors switch between nM and ug/ml for BASA, and it’s hard to follow what the actual concentration is. This makes comparisons among the tables and figures harder than it needs to be. As materials can have different masses, use of molarities is most precise and strongly encouraged.
  2. I could not find where the authors indicated where the BASA they used came from: was it purchased? If so, where? Synthesized? How? This should be documented as carefully as everything else.

Author Response

1. The authors switch between nM and ug/ml for BASA, and it’s hard to follow what the actual concentration is. This makes comparisons among the tables and figures harder than it needs to be.

Response: For all the molecular analysis molar terms were used, and this was the predominant unit throughout the manuscript. We used ug/ml for the microbiological and cell based assays as this is conventional for these studies. The MIC for BASA was >1 ug/ml, which is comparable potency to other promising antibiotic in development.

2. I could not find where the authors indicated where the BASA they used came from: was it purchased? If so, where? Synthesized? How? This should be documented as carefully as everything else.

Response: BASA was synthesised in our laboratory. This has been described previously in Ref 7. We have added an additional sentence in the Introduction stating "A full description of its chemical synthesis and characterization has already been described [7]."